# Listen to the Brain–Auditory Sound Source Localization in Neuromorphic Computing Architectures

**DOI:** 10.3390/s23094451

**Published:** 2023-05-02

**Authors:** Daniel Schmid, Timo Oess, Heiko Neumann

**Affiliations:** 1Institute of Neural Information Processing, Ulm University, James-Franck-Ring, 89081 Ulm, Germany; heiko.neumann@uni-ulm.de; 2Bernstein Center Freiburg, University of Freiburg, Hansastr. 9a, 79104 Freiburg im Breisgau, Germany; timo.oess@bcf.uni-freiburg.de

**Keywords:** neuromorphic computing, neuromorphic hardware, TrueNorth, SpiNNaker, sound source localization, interaural level difference, lateral superior olive

## Abstract

Conventional processing of sensory input often relies on uniform sampling leading to redundant information and unnecessary resource consumption throughout the entire processing pipeline. Neuromorphic computing challenges these conventions by mimicking biology and employing distributed event-based hardware. Based on the task of lateral auditory sound source localization (SSL), we propose a generic approach to map biologically inspired neural networks to neuromorphic hardware. First, we model the neural mechanisms of SSL based on the interaural level difference (ILD). Afterward, we identify generic computational motifs within the model and transform them into spike-based components. A hardware-specific step then implements them on neuromorphic hardware. We exemplify our approach by mapping the neural SSL model onto two platforms, namely the IBM TrueNorth Neurosynaptic System and SpiNNaker. Both implementations have been tested on synthetic and real-world data in terms of neural tunings and readout characteristics. For synthetic stimuli, both implementations provide a perfect readout (100% accuracy). Preliminary real-world experiments yield accuracies of 78% (TrueNorth) and 13% (SpiNNaker), RMSEs of $41^{\circ }$ and $39^{\circ }$, and MAEs of $18^{\circ }$ and $29^{\circ }$, respectively. Overall, the proposed mapping approach allows for the successful implementation of the same SSL model on two different neuromorphic architectures paving the way toward more hardware-independent neural SSL.

## 1. Introduction

### 1.1. Motivation

Audition plays an important role in survival as it helps us to sense our environment omnidirectionally and localize sound sources precisely. Such localization is not trivial since perceived sound signals do not explicitly convey localization cues. Instead, these cues need to be computed from differences in the incoming sounds of the left and right ear. In the case of lateral localization, two binaural cues are computed: the interaural time difference (ITD) and the interaural level difference (ILD) [1]. As the name implies, the ITD is based on the difference in the arrival time of sound signals on one ear compared to the other ear, and its encoding in mammals is based on a sophisticated inhibition circuit [2]. The interaural level difference (ILD) is the other cue for lateral sound source localization (SSL). ILD cues are primarily used for high-frequency sounds (>1500 Hz [3]) since high frequencies are attenuated by the head and, therefore, create a difference in the intensity levels between ipsi- and contralateral side (Figure 1A). The encoding of these cues takes place in the lateral superior olive (LSO) [4]. By a weighted combination of excitatory ipsilateral and inhibitory contralateral inputs single LSO neurons exhibit tuning curves that relate their sensory stream inputs monotonously to ILD values (see Figure 1B for a typical tuning curve).

Beyond providing an understanding of brain function, models of ITD and ILD computation can be used for real-world applications of biologically inspired SSL. For such applications, biological inspiration can not only be taken on an algorithmic level. Moreover, the employed hardware can be inspired by principles from biology in terms of neuromorphic computing paradigms. Neuromorphic hardware platforms provide specialized computer architectures that mirror the structure and function of neurons and their interaction in networks [5,6,7]. Such hardware comes with benefits in terms of real-time capabilities and energy efficiency. These are achieved by adhering to event-based processing principles instead of sampling and processing sensory information temporally uniformly.

So far, there is no generally agreed-upon neuromorphic hardware architecture, but many different kinds exist, each with its own strengths and weaknesses, and use cases [6,8,9]. Likewise, improving the energy efficiency of neuromorphic systems, and hence developing new hardware, is still an ongoing effort [10,11,12]. As an implication, also software frameworks and available components to implement neuron models vary broadly. Developing even one neural network for one architecture can, thus, already be an endeavor. While some approaches of SSL have been brought to neuromorphic platforms already, efforts to allow for utilizing the same SSL model on multiple different such platforms are still lacking. Here, we propose a mapping procedure to allow for using the same underlying biologically inspired SSL model across different neuromorphic platforms. Following the mapping, procedure we implement the same SSL model on two different neuromorphic platforms and compare the results between both implementations.

**Figure 1 sensors-23-04451-f001:**
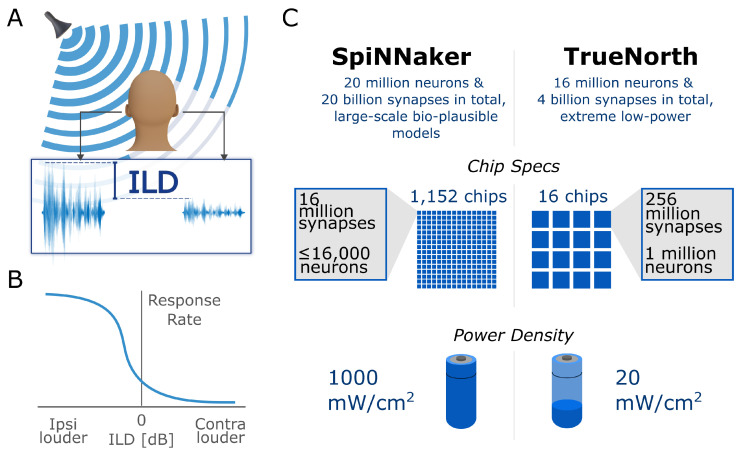
Overview of relevant concepts for neuromorphic sound source localization. (**A**) Principle of interaural level difference (ILD) computation. High-frequency sounds coming from the ipsilateral side are attenuated by the head and, therefore, have a smaller amplitude at the contralateral ear. This difference in amplitude, or sound level, between the ipsi- and contralateral ear varies systematically with the azimuth angle α and is encoded in the interaural level difference in lateral superior olive (LSO) neurons. (**B**) Typical LSO neuron response over different ipsi- and contralateral sound levels. Such a response curve varies for different LSO neurons, i.e., it shifts along the x-axis and the steepness varies. (**C**) Comparison of hardware specifications for the neuromorphic chips used in our approach (comparison based on [13]).

### 1.2. Related Works

Many proposals of biologically inspired SSL exist already, yet only a few of them have been brought to neuromorphic hardware (Table 1). While we focus on these neuromorphic biologically inspired SSL approaches here, broader overviews concerning SSL in general or with applications to robotics can be found in Desai and Mehendale [14] and Rascon and Meza [15], respectively.

Studying biologically inspired SSL in neuromorphic hardware, so far, is mostly concerned with processing ITD cues and implementations tailored toward specific hardware. Moreover, testing of the systems varies widely in terms of input data. A very early implementation of Lazzaro and Mead [16] built an analog integrated circuit (IC) and tested the linear relationship between neuron responses and input ITD with simple click stimuli fed to the board. Glackin et al. [17] developed their spiking neural network based on a model of the medial superior olive (MSO) at first on a conventional computer and then brought it to field programmable gate array (FPGA) based hardware to accelerate the system’s performance. The model has been trained using spike-timing-dependent plasticity (STDP) based on ear canal recordings of a domestic cat acquired in a sound-dampened chamber. Similarly, Xu et al. [18] utilized a correlational approach based on ITD-related onset-timing features extracted from an auditory filter device mimicking cochlear processing. Using sound samples from speakers in a reverberant environment the extracted features have been used in either a regression or extreme learning machine (ELM) approach for SSL. Escudero et al. [19] developed an FPGA-based sound tracking system, which turned a robotic platform towards the sound source. Notably, the system relied on ILD cues. It was tested with pure tones recorded in a classroom and ILD cues have been used in a control loop after an initial calibration phase per scenario. A recent investigation by Schoepe et al. [20] combined an FPGA and a SpiNNaker board to establish a closed-loop robotic control platform. The system capacity was tested in a real-world setting with pure tones and human speech played back from a loudspeaker. Additional components of time-to-rate, ring attractor, and center detector networks transform ITD-based encodings from a time-difference encoder (TDE) to motor commands. The TDE computation itself was not conducted on SpiNNaker, but on the FPGA board to meet the required timing constraints of the ILD-based approach. Oess et al. [21] first reported a model of ILD-based SSL on the TrueNorth NeuroSynaptic System, which we subsume and extend in our current communication. The system was tested on synthetic data and recordings of natural sounds played back from a speaker in a sound-dampened chamber. Despite these approaches on either generic (ICs, FPGAs) or specific neuromorphic (SpiNNaker, TrueNorth) hardware, lately, also novel biologically inspired resistive random access memory (RRAM) approaches have been proposed. So far, these provided proof-of-concept results based on ITD cues and controlled laboratory setups [22,23].

Taken together, biologically inspired SSL approaches on neuromorphic hardware, so far, focused on either specific hardware platforms or rather freely definable FPGA implementations. Notably, most of the approaches followed the route of ITD-based SSL and tests were mostly restricted to laboratory or indoor settings and synthetic or played-back sounds.

### 1.3. Summary

Here, we propose a generic approach for how to map biologically inspired neural networks to different neuromorphic platforms and, subsequently, apply it to a binaural SSL model, which encodes ILD values for the horizontal plane.

The employed neural architecture yields a mechanistic explanation of the LSO and, hence, gives rise to a biologically inspired algorithm of SSL. It is based on single-compartment conductance-based neuron models, which serve as computational units for signal integration and generation of output responses. The activations of single units in the model represent the average membrane potential of a group of neurons calculated as the sum of excitatory, inhibitory, and leak conductances. Their dynamics are described by gradual activation dynamics of a neuronal population [24].

These dynamics can be simulated by numerically integrating systems of ordinary differential equations [25]. However, such an approach is neither very cost- nor time-efficient to compute. To circumvent these computational issues and to account for the biological principles of neural information processing in real neurons, we adopt an event-based, i.e., spiking, representation of responses. Responses are generated asynchronously based on an integrate-and-fire principle  [26].

To make full use of the event-based model paradigm, we implement the model on neuromorphic platforms. Importantly, since no generally agreed-upon platform exists yet, we here propose a generic approach for mapping such models to different platforms coming from an original rate-based model description. As a first step of the mapping, we identify generic computational motifs within the original rate-based model. During a second step, a transformation of this generic component onto patterns of spiking response characteristics in neuromorphic hardware is performed. While the first two steps are platform independent, we suggest a consecutive step that operationalizes the mapping onto specific brain-inspired hardware platforms.

We exemplify this mapping based on two neuromorphic platforms, namely the TrueNorth Neurosynaptic System [5,27] and SpiNNaker [28] (Figure 1C). Both systems are designed to utilize digital computing technology relying on a CMOS hardware process. TrueNorth has been developed by IBM as part of DARPA’s SyNAPSE project. The architecture is functionally organized as a network of neurosynaptic cores, each one defining a canonical cortical microcircuit [5]. TrueNorth’s computation is extremely energy efficient, consuming 70 mW of power in operation. This is achieved by asynchronous event-driven neuron activation, hardware implementation of specific neuron models, and an on-chip network realization that interconnects all neurosynaptic cores avoiding an off-chip memory component. A single chip contains one million neurons with 256 million synapses. The system is delivered as 1-, 4-, and 16-chip board architecture [13,27]. The SpiNNaker network architecture utilizes a general-purpose parallel computing system that can run neurons of different specifications in software. It has been developed at Manchester University within the EU Flagship Human Brain Project (HBP). SpiNNaker is a many-core architecture utilizing small integer cores and incorporating a communication framework that is optimized to send large numbers of very small data packages (conveying neural spikes) to many destinations following a multi-cast principle [6,29]. Delivering spike activations to arbitrary receiver neurons is based upon a packet-switched Address Event Representation (AER). A SpiNNaker node contains 18 ARM processor cores and consumes 1 W power for a fully loaded 18-core package. The system is delivered in two circuit board configurations, namely a 4-node (72-core) and a 48-node (864-core) board.

We present on-chip model simulations for the defined neural network mechanisms for ILD-based auditory SSL computations of synthetic and naturalistic sounds. The model has been successfully implemented on the different hardware designs of TrueNorth and the SpiNNaker neuromorphic architectures, respectively, based on the proposed generic mapping approach. We demonstrate similar performance in the response characteristics of the resulting neuromorphic algorithms implemented on the different platforms. These model implementations realize model circuits towards a real-time and energy-efficient SSL for real-world applications.

## 2. Materials and Methods

Here, we describe a flexible mapping approach that deploys the same model and functionality of auditory sound source localization to different neuromorphic frameworks. With such mapping an operationalization across neuromorphic hardware can be realized, despite framework-specific differences (Figure 2).

### 2.1. Generic Neuron Model to Build Complex Network Function

The strength of the approach is rooted in describing nodes in rate-based neural network models by means of a generic neuron model. This neuron model is inspired by the brain’s function and architecture. It abstractly captures the canonical computational principles of local microcircuits as found in the brain and has been investigated thoroughly with respect to its computational properties [30]. Each such neuron is described by a first-order ordinary differential equation (ODE) and exhibits several desirable properties. It is classified as conductance-based, i.e., the temporal evolution of integrating inputs into the neuron’s state variable (membrane potential) adheres to the specific dynamics. This integration of inputs naturally renders the model’s operating range bounded, preventing an explosion of state values. By altering its parameters, the model has steerable characteristics, such as altered temporal smoothing response properties (via a time constant) and response selectivity (via non-linear activation functions and space-feature kernel tunings). Additionally, neurons can be connected by excitatory or inhibitory connections, which impact state values in opposite directions (increasing for excitatory and decreasing for inhibitory connections, respectively). By balancing off these connection types, the model is further stabilized in its computation on a network level. Capitalizing on the understanding of such single neurons, it becomes manageable to build neural computational motifs, such as difference computation, or comparison operations, in a modular fashion. Likewise, building bigger network structures out of them, to investigate more complex functions, for example, auditory SSL in the brain [31] or visual contrast detection [32] becomes feasible.

### 2.2. Generic Mapping Approach to Neuromorphic Hardware

Our generic mapping procedure subsumes and generalizes previous work of deploying the SSL model to TrueNorth [21] and is showcased by likewise paving the way to a functioning SpiNNaker implementation. The mapping procedure starts with generic, hardware-agnostic steps and requires successively more and more hardware-specific ones (Figure 2). Starting from a rate-based model, generic design decisions can still take place in the rate-based, continuous paradigm and can happen on two different levels of neural model description. On a network-level, the required computational motifs, based on node interconnectivity and connection type, are identified. On a node-level, the required components and properties in terms of the canonical neuron model are determined.

After determining connectivity patterns and neuron model properties, a first mapping takes place for each of them. This step is rather generic in nature but might already be informed by limitations of the target neuromorphic hardware and framework, such as limited resolution for representing values. On the network-level, connectivity patterns need to be described in terms of connection matrices, or kernels. Dependent on the respective framework’s demands, the kernel values might need to be quantized. On the node-level, the neuron model needs to be mapped from its canonical rate-based form into a spike-based version. The spike-based version temporally integrates incoming spikes to feed them into the state value ODE and outputs time-discrete, binary spikes based on the respective state value once again. Here, multiple options exist in how the exact spike-integration and -generation mechanisms might be chosen. Dependent on the target framework additional properties might be required, such as (non-)stochastic generation mechanisms or simplified temporal update rules. Up to this point, it is still possible to simulate the model on conventional hardware by using a spike-based neuron model framework, such as NEST [33].

Subsequently, a hardware-specific mapping step follows on both description levels by implementing the model within the respective framework. This step comes with the necessity of access to framework-specific knowledge about API usage, software setup, and dependency management. On a network-, as well as node-, level this might include describing the components in terms of the provided API methods or classes. This step might come with further model abstractions, the requirement to build upon pre-existing software components, and running routing processes to make for the proper placement of nodes and connections among logical units of the respective architecture. Likewise, interfaces for providing the required input to the network might need to be implemented or configured.

Ultimately, the implemented models can be executed on the respective neuromorphic hardware. There, parameters and values can be fine-tuned to best match the desired functional response properties as originally investigated in the rate-based paradigm.

### 2.3. Mapping the Auditory Sound Source Localization Model

Here, we describe how the generic mapping approach applies to a model of auditory sound source localization (SSL). The SSL model of interest was first proposed in Oess et al. [31] for how sound source information from two ears can be combined in a biologically plausible fashion, and later extended into a network for being able to predict intensity level differences (ILDs), and hence azimuth angles, and implemented on the TrueNorth architecture by Oess et al. [21].

We first outline the key features of the model to provide the reader with an intuition about its working principles. Afterward, we describe how the proposed mapping procedure applies to the model. Further mathematical details about the LSO model and its SSL extension can be found in the respective original publications [21,31].

#### 2.3.1. The Auditory Sound Source Localization Model

The SSL model is based on neurons that mimic the response properties of lateral superior olive (LSO) and medial nucleus of the trapezoid body (MNTB) neurons, which are causally involved in mammalian SSL (Figure 3, center). A set of LSO and MNTB neurons exists in each hemisphere with connections to sensory input from both ears. They jointly compute ILD values per input frequency. For each frequency ω, different neurons exhibit different sensitivities *p* correlated with azimuth angles (Figure 3, inset). Neurons of both types, LSO and MNTB, are described by a conductance-based neuron model in the form of an ODE: (1)$$\tau_{r}{\dot{r}}_{\omega ,p}=-\alpha_{r}r_{\omega ,p}+\left(\beta_{r}-r_{\omega ,p}\right)\cdot \sum_{\omega^{\prime }}I_{\omega^{\prime }}^{Exc}\Lambda_{\omega \omega^{\prime },p}^{Exc}-\left(\gamma_{r}+\kappa_{r}r_{\omega ,p}\right)\cdot \sum_{\omega^{\prime }}I_{\omega^{\prime }}^{Inh}\Lambda_{\omega \omega^{\prime },p}^{Inh}$$
where $\dot{r}$ is the temporal derivative of the membrane potential of neuron *r* and $\tau_{r}$ defines the time constant of its temporal evolution. The change of *r* depends on a leak-term (right, first term) with constant $\alpha_{r}$, which drives the neuron back towards its resting potential of 0. This stabilizes the dynamics and ensures that only recent information is represented by the neuron. Input $I_{\omega^{\prime }}^{Exc}$ excites the neuron driving it towards positive values (second term), while input $I_{\omega^{\prime }}^{Inh}$ inhibits the neuron driving it towards, negative values, respectively (third term). The operating range of the neuron is thereby bounded by reversal potentials of $\beta_{r}$ towards positive values and $\gamma_{r}$ towards negative values, respectively, further ensuring a stable operating regime. Parameter $\kappa_{r}$ allows for different inhibitory computations of either divisive or subtractive kind, for larger or smaller parameter values, respectively. Excitatory, as well as inhibitory, inputs are summed across neighboring frequencies $\omega^{\prime }$ and weighted by a connection weight from kernels $\Lambda_{\omega \omega^{\prime },p}$. Together, the kernels determine the model’s connectivity structure (Figure 3, center). An output rate per neuron is determined via a non-linear activation function $g_{r}\left(r\right)$. Parameters and kernels are specific to the respective neuron type.

By using this excitatory-inhibitory interaction in LSO neurons, neural tunings correlate with ILD values (Figure 3, inset). Together, the *p* differently tuned neurons form a representation from which a readout population can determine a location estimate in terms of the azimuth angle. This readout happens per frequency ω. The readout population is based on Oess et al. [21] and consists of simple neurons of form
(2)$$u_{\omega }=\sum_{p}g_{r}\left(r_{\omega ,p}\right)k_{p},$$
where a response *u* is determined by the weighted summation across LSO neurons. Estimates are formed per frequency band ω. The readout neurons’ response rates encode their estimates in the target domain. "rates to rate" Identical weights $k_{p}$ are applied to each frequency band ω. They are determined from a single frequency band by linear regression over a set of training examples containing target domain values $u^{★}$ [21]. Here, boldface notation demarcates vectors against scalars.

From this discrete set of estimates $u_{\omega }$, a final azimuth estimate $\hat{\theta }$ can be computed by a procedure
(3)$$\hat{\theta }=f\left(u\right),$$
where *u* is the set over all $u_{\omega }$. Following Oess et al. [21], *f* consists of the following steps: At first, per frequency band ω, evidence for the different target domain values is assigned. To this end, the estimate $u_{\omega }$ is compared against the discrete set of target values from the training examples $u^{★}$ via a difference computation. The differences are scaled by a mismatch penalty constant *c* and the most likely candidates are then determined by thresholding the evidence, i.e., $max(1-c\cdot \left(u_{\omega }-u^{★}\right),0)$, where 1 would yield a perfect match. A final estimate is then computed by summing up the evidence across frequency bands ω and choosing the encoded value of the maximum, i.e., the target value for which the highest evidence was accumulated.

#### 2.3.2. Application of the Mapping Procedure

In the present study, we focus on mapping the core components of the model, namely the MNTB-LSO circuit. Notably, the mapping is not restricted to only this part of the overall network and could likewise be applied to the ILD readout mechanism.

To map the SSL model to TrueNorth or SpiNNaker, at first, the model’s connectivity motifs and neuron properties need to be identified. On a connectivity level, LSO neurons receive direct excitatory input from the ipsilateral input and inhibitory input from the contra-lateral input via an indirection of the MNTB neurons. These connections obey a one-to-one-connectivity pattern. The excitatory-inhibitory pairing of ipsi- and contra-lateral inputs yields a local difference computation mechanism. On a neuron level, LSO neurons perform a difference computation by means of their excitatory and inhibitory conductance types (W-S nodes, for weighted summation). MNTB neurons only receive excitatory input from the contra-lateral site and project it further to the ipsilateral site serving the function of a relay station.

Next, the identified motifs are simplified, and model neurons are described in spiking terms. While the MNTB neurons add to biological plausibility, they do not impact the computation on a network level (given their one-to-one connectivity pattern). Hence, they can be subsumed within a direct connection of contra-lateral input to the ipsi-lateral target LSO neuron’s inhibitory conductance input. Sparing out this indirection means, likewise, to spare out spending dedicated MNTB neuron models in the circuit. Additionally, connection kernels described in terms of Gaussians can be simplified and quantized by expressing them as binomial filter weights. This leads to four required connections (excitatory ipsilateral and inhibitory contra-lateral per hemisphere), which are each described by a binomial kernel. Introducing an additional set of relay neurons (S-T nodes for spatiotemporal smoothing) allows for reducing this multiplicity of four kernel-based connections to two (one spatial smoothing kernel per hemisphere) and four one-to-one connections (excitatory ipsilateral and inhibitory contra-lateral per hemisphere; Figure 3). Kernelized computations cannot directly be expressed in terms of convolutions on neuromorphic hardware, as neurons and computation are distributed across many chips and cores. Thus, reducing the number of connections requiring kernels will significantly reduce the number of connections to be instantiated in hardware later on. Different W-S nodes weigh ipsi- and contra-lateral inputs differently. Their connectivity structure is not further reducible and will be kept as is. This leads to scalar weights of {1/(p+1),⋯,p/(p+1)} for positive, ipsi-lateral connections and {p/(p+1),⋯,1/(p+1)} for negative contra-lateral connections for the set of *p* W-S nodes per frequency band ω.

To map the rate-based neuron description to a spike-based one, several viable options exist. Here, we chose a rather common one by describing them in terms of leaky-integrate-and-fire-like neurons. More specifically, we chose an adaptive exponential firing (AdEx) type version, which has been shown to function well with the canonical model description in the past [34]. In fact, this means keeping the basic conductance-based ODE structure (Equation (Equation 1)) intact, replacing the non-linear activation function output $g_{r}$ by a constant threshold and extending it for additional components. A compare-and-reset mechanism compares the state value against a threshold value, elicits an output spike, and resets the state value if the threshold was crossed. An additional adaptation component extends the neuron’s state for another component and allows it to exhibit desirable behavior adaptive on the input spike rate [35]. To be utilizable in the state value computation, incoming input values are aggregated into excitatory and inhibitory conductance values and converted from a discrete spike representation into a continuous one. This is achieved by dedicated synapse dynamics, implementing an exponential moving temporal averaging process over each input spike stream (cf. [36]). This spike-based neuron model description can be employed for S-T, as well as W-S neurons.

After this point, a hardware-specific mapping must be pursued. For TrueNorth, details have been reported previously [21]. In a nutshell, binomial filter weights need to be expressed within a specific range of integer values coming with a respective approximation, and the spike-based neuron model needs to be approximated and implemented by the existing hardware neuron type and its options for parametrizations. For the SpiNNaker implementation, the described connections must be expressed by SpiNNaker’s version of the pyNN framework [37], with the existing types, such as OneToOneConnectors, or ArrayConnectors encapsulating kernel weights within StaticSynapse object weights. The neuron model can be mapped rather straightforwardly, by implementing a new type of SpiNNaker neurons in the framework’s Python and C code base. (See https://spinnakermanchester.github.io/spynnaker/6.0.0/NewNeuronModels6.0-LabManual.pdf (accessed on 6 March 2023)) for an introduction. Additionally, model inputs can be described by SpikeSourcePoisson or SpikeSourceArray objects. The overall model can then be expressed in Python code, from which the SpiNNaker framework is controlled, and simulations can be started if access to the hardware exists. Similarly, to the rate-based neuron model [21,31], model parameters (Equation (Equation 1)) have been determined manually, while readout weights $k_{p}$ (Equation (Equation 2)) have been determined by linear regression over a set of input data. As our focus lies on the part of the model, which performs ILD computation, we chose to keep the estimation procedure from the readout population as simple as possible in case of the SpiNNaker implementation (i.e., no mismatch penalty c=1 and no thresholding against 0).

The SpiNNaker implementation consists of 1024 neurons running in real-time, exactly like reported for the TrueNorth implementation [21]: 64 S-T neurons per hemisphere and 64×7 W-S neurons per hemisphere for 64 frequency bands ω and 7 different ILD tunings *p*. So far, the implementation specifies placing maximally 65 neurons per core (via the set_number_of_neurons_per_core method) leading to a total of 16 out of 864 cores occupied by the model itself. The maximal neuron number per core was chosen conservatively to allow for safe model execution and could likely be optimized further.

### 2.4. Experiments for Model Comparison

We compare the implementations of the auditory SSL model on TrueNorth and SpiNNaker against each other in two experiments following the procedures in [21]. The first experiment serves to verify that the model can encode ILD values reliably, and the second experiment tests its real-world capabilities. While the presented SpiNNaker data were acquired as part of this work, we base our analysis of the TrueNorth implementation on the previously acquired data of [21]. The SpiNNaker model has been implemented in Spinnaker 6.0.0 and executed on a 48-node board.

The first experiment tests the viability of the implementation on both neuromorphic architectures for whether it can reliably predict ILD values, and thus, sound source locations. To this end, synthetic input of different ILD values is provided mimicking sound source position at different azimuth angles. ILD values are varied in the range of −1.0 to 1.0 in steps of 0.1. In this synthetic setting, only a single frequency band receives input and is evaluated. Across the range of input values, the neural responses of W-S units are visualized to better understand how they correlate with the input, i.e., their tuning properties. Combining the W-S units of both hemispheres by the readout mechanism allows one to inspect how much information about the ILD values is encoded within their representation. The readout weights $k_{p}$ of the models are fit by regressing the system of linear equations given by the W-S unit outputs and the target ILD values. Based on the determined weights, ILD predictions for the dataset are evaluated for how well they can describe the data. Therefore, for each model, evidence values from the final readout stage before choosing the maximum are plotted against the true ILD values. Finally, the accuracy of each model is computed from the predicted ILD values of the readout stage.

In the second experiment, the models are evaluated within a real-world setting to estimate their performance for everyday sound sources. To achieve plausible physical properties, two microphones have been placed in the ear canals of a 3-D printed human head with human-like shaped ears. The head was rotated with angles of ($-90^{\circ }$ to $90^{\circ }$ in steps of $10^{\circ }$). A stationary sound source was positioned at a defined distance. The elevation was fixed to $0^{\circ }$ leading to changes in the azimuthal plane only (setup comparable to Figure 1A). For each angle, recordings of eight different natural sounds have been acquired. Recordings have been conducted in a sound-attenuated chamber. Subsequently, the recorded sounds have been preprocessed by a gammatone filter bank to mimic human hearing. Finally, the data has been converted to a spike-based representation, to be used as input for the neuromorphic hardware by generating spikes along the recording duration with a probability based on the intensity of each frequency. The recorded data were originally reported in [21]. As for the TrueNorth chip, we adapt the spike rate to match the operating range for the model’s input on SpiNNaker. To this end, we subsample the generated spikes by a factor of 10. The spike data are fed to on-chip simulations of the SpiNNaker model and the output spike trains of the model’s W-S units are recorded for further analysis. A rate estimate per W-S neuron is formed by computing the mean across the last half of the spike train’s duration to exclude potential onset effects in the analysis. Next, weights of the readout population are fit by regression over the different angular recordings for W-S units of a single frequency band (no. 32 out of 64) for one of the eight sound types (white noise). This single set of weights is then used across all frequency channels and all sound types to calculate azimuth estimates from the readout population, respectively. Different to the protocol described in Oess et al. [21], we do not exclude any part of the frequency spectrum for calculating the estimates. Therefore, the readout is solely based on the linear weighting of W-S units per frequency channel, summation across frequencies, and choosing the azimuth value for which the highest evidence was accumulated.

The azimuth estimates acquired for SpiNNaker, and those for TrueNorth from [21], are then analyzed for their accuracy, mean absolute error (MAE) and root mean squared error (RMSE) against the known ground truth azimuth. We adhere to the protocol of Oess et al. [21], rescale the respective estimates per sound type, and exclude non-informative sound sources from the analysis. In this case, we restrict ourselves to the subset of the four sound types. The accuracy is calculated again as for the first experiment.

## 3. Results

In the first experiment, one finds for both models a characteristic of rising activities for greater ILD values, and hence sound source azimuth, towards the respective ipsilateral side (Figure 4a,b). Combining both hemispheres in the model leads to a mirrored sensitivity profile, where sensitivities of corresponding channels reach the same levels for ILDs of zero, i.e., for stimuli presented straight ahead of the auditory observer. Furthermore, tuning curves are qualitatively replicated across the channels of W-S units of a single frequency, although they differ in the weighting of ipsi- versus contra-lateral inputs. Depending on the weighting, their curves are shifted in their selectivity, becoming only responsive for increasingly larger ILD values. While the TrueNorth and SpiNNaker tuning curves differ in their exact shape, the models’ responses are in good qualitative agreement and both replicate the basic response properties of the underlying rate-based LSO model [31].

Both models reliably encode the whole range of ILD values for synthetic input stimuli via their respective readout population (Figure 4c,d). Dependent on whether a mismatch penalty c and thresholding against 0 are used or not as part of the readout procedure *f* (cf. Equation (Equation 3)), the encoding becomes either more localized (as in the case of the TrueNorth model) or gradual across ILD neighborhoods (as for the SpiNNaker model). Importantly, though, choosing the maximally activated readout neuron per input recovers the correct ILD in both implementations. Both implementations achieve a perfect score of 100% indicating that ILD information can linearly be decoded from W-S neuron activities. This can also be seen from the main diagonal of the population readouts across the range of ILDs (Figure 4c,d). There, for both models, the diagonal holds the highest neuron activities per respective column (i.e., input sample). As the chosen readout mechanism is quite simplistic (weighted summation, optional bias, and non-linearity, averaging across frequency channels, and choosing the maximally active neuron), it can be concluded that both models are well able to encode sound source azimuths from ILDs.

In the second experiment, the computed accuracy across the dataset is 78% for TrueNorth and 13% for SpiNNaker at $10^{\circ }$ resolution. MAEs are determined to $18^{\circ }$ and $29^{\circ }$, and RMSEs to $41^{\circ }$ and $39^{\circ }$ for TrueNorth and SpiNNaker, respectively, for normalized azimuth values in the range of $-90^{\circ }$ to $90^{\circ }$ (Figure 5). The general trend (mean) and variation (error margins) of azimuth estimates can as well be seen in the graphical data (Figure 5, blue curves). They indicate aggregates over the different sound types per tested angle. Comparing the estimates against the correct azimuth values, both implementations follow the general trend of the ground truth (Figure 5, orange curve). TrueNorth matches the ground truth more closely for most of the samples, as also indicated by the higher accuracy and lower MAE. However, both implementations are comparable in terms of RMSE, as the SpiNNaker estimates vary more uniformly across recorded angles with less strong outliers.

## 4. Discussion

In this paper, we propose a generic mapping approach to implement neural mechanisms for sound source localization (SSL) in the horizontal plane on multiple neuromorphic platforms. The SSL model utilizes the differences in sound level of the source signal at the two ears, or microphones, corresponding to the interaural level difference (ILD) principle. The main aim of this communication article is to propose a principled approach to map the underlying neural mechanism onto a target neuromorphic platform for the neuromorphic implementation of the core functionality. We exemplify the mapping procedure from two different platforms, namely IBM’s TrueNorth neurosynaptic chip (developed in the DARPA Synapse program; [5,27]) and the SpiNNaker platform (developed in the EU Flagship Human Brain Project, HBP; [28]). We emphasized that the first step in this mapping cascade is intrinsically related to the neural model, while the last part is specifically designed to adopt the properties of the neuromorphic target architecture. During the first mapping step, connectivity motifs are identified and neurons of the original model are characterized in terms of a generic neuron model [30]. Next, the connectivity motifs and neurons are mapped onto neuromorphic-compatible simplified connection patterns and spike-based neurons [35]. After these generic steps, a final step specific to the target platform takes place, during which connections and neurons are formalized within the target neuromorphic framework and parameters are fine-tuned. Using the mapping procedure to implement the neural SSL model on both platforms, TrueNorth and SpiNNaker, respectively, we have conducted two main experiments to demonstrate the functionality as a proof of concept.

On synthetically generated input, we validated that the neuromorphic implementations of the SSL model for both architectures are well capable to encode ILD values, and hence azimuth location, across the complete range of inputs. The input–output behavior of the model was investigated in this experiment for a single frequency band. Computing the accuracy of the readout population resulted in a perfect score of 100% for both implementations indicating that ILD information can be decoded linearly from the model’s W-S units. This validates the feasibility of the proposed mapping procedure and, in line with previous findings [21], the functionality of the SSL model.

During a second experiment, we tested the real-world capabilities of the mapped models. Both models showed similar qualitative behavior under real-world input in providing azimuth estimates that follow the overall course of calculated ground truth values. Quantitatively, the performances in the real-world setting between the TrueNorth and SpiNNaker implementation differ for accuracy (78% vs. 13%) and MAE ($18^{\circ }$ vs. $29^{\circ }$), while being more similar in terms of RMSE ($41^{\circ }$ vs. $39^{\circ }$). The discrepancy might be attributable to multiple sources of influence. While in the synthetic setting performances between both models were well aligned, differences only became apparent in the real-world experiment. For these input types, the SpiNNaker experimental protocol deviated slightly from the TrueNorth one, providing one possible reason. For the second architecture, i.e., SpiNNaker, the results of the second experiment are to be interpreted as a first proof-of-concept, as we restricted the mapping approach to the SSL model’s core ILD mechanism. As an implication, we deliberately chose to not perform most of the proposed additional pre- and postprocessing steps (cf. [21]). On a post-processing side, this means that we did not exclude any poorly performing frequency bands from forming the azimuth estimate. On a pre-processing side, this means, that no time-bin-wise cross-normalization of left and right input was performed. Thus, the quantitative results can be seen as lower bound performance estimates for the given experiments, which nevertheless capture the essentials of biologically inspired SSL. While quantitative performance measurements are better for the TrueNorth data in terms of accuracy, a visual comparison of mean azimuth estimates between the architectures and the ground truth estimates reveals similar qualitative traits. Both implementations follow the general trend and desired behavior of the ground truth’s ILD response curve. We are confident, that applying such pre- and postprocessing techniques likewise to the SpiNNaker model will lower or even close the quantitative gap.

As the basic underlying SSL mechanism is well captured in both architectures, these quantitative discrepancies are rather suggestive of an avenue towards future development directions, than revealing shortcomings of the proposed mapping. Namely, to overcome the current dependency of the employed SSL model on pre- and post-processing of the data i.e., cross-normalization of input ranges, and selective admission of frequency bands into the ILD estimate formation). These processing steps strongly reduce confounders to the ILD estimate stemming from differences in overall loudness (cross-normalization step) and variability of frequency bands dependent on sound source type (frequency admission step). A remedy to these situation-dependent effects lies in making the model adaptive to context. Such context-adaptivity lends itself perfectly for an all-neuronal implementation within the already utilized canonical neuron model. For example, eliminating absolute level information while retaining contrasting one between computational elements (as required for the cross-normalization step) has been shown in the context of biologically plausible visual information processing of oriented contrast information [38]. Likewise, context-dependent selection of information (as required for the frequency admission step) is a prime example of top-down attention in the context of visual object detection, where the visual system needs to selectively filter its input for features matching those of a higher-level aggregate representation of the object [39]. Indeed, both functionalities have been studied previously using the canonical neuron model that comprises as well the mapped SSL model [30,40]. In such context-adaptability lies the advantage of using neuronal models over conventional approaches. By way of the dynamic computation of the neurons employed, they can be naturally extended to display such adaptive behavior (cf. [31] for the case of history-dependent adaptation of ILD selectivity). Currently, the neuromorphic model implementations occupy only a fraction of the overall available compute resources, for example, 16 out of 864 cores for SpiNNaker). Thus, the implementation would offer enough space for further model components and adaptations, even if the additional resource requirements for input handling and monitoring are taken into account.

As part of this communications article, we aimed to exemplify the proposed mapping procedure and compared the SSL model implementations on TrueNorth and SpiNNaker. To make for a proper comparison, the same experiments were performed with the new SpiNNaker implementation as previously reported for TrueNorth [21]. This set of experiments served our purpose well by providing a common ground to compare both implementations. However, it has several limitations when it comes to capturing the variability of the real world. The synthetic data were restricted to a single frequency band. On the one hand, this allowed for precise control and investigation of the neural tuning properties. On the other hand, it lacks the variability and bandwidth of real-world sounds. The real-world experiments were based on a dataset first reported by Oess et al. [21]. The data were acquired in a sound-dampened chamber and provided a single stationary sound source per recording from a limited set of sound types. Thus, to draw stronger conclusions about the performance of the neural SSL model itself, further experiments will be needed, on either platform, under more demanding conditions. For instance, synthetic data could be generated comprised of more than one frequency band and defined temporal variations. Likewise, real-world recordings under realistic conditions would further test the model. These could consist of multiple sound sources from different angles at once recorded in a reverberating environment. The model on which we exemplified the mapping adheres to principles of biological auditory processing of ILD. Thus, it will underlie uncertainties and ambiguities, such as the cone of confusion [41]. Incorporating further acoustic features, such as ITD or spectral cues, would be interesting extensions to the model. This way, model performance might be increased and extended to estimates along the elevation angle as well. Likewise, replacing the readout procedure (Equation (Equation 3)) with a deep neural network could harness the model for technical applications and result in more precise location estimates [42].

## 5. Conclusions

In the present study, we outlined a generic approach for mapping rate-based neural architectures to spike-based neuromorphic hardware applied to the case of auditory sound source localization (SSL). We investigated the functional equivalence of the mapping for two neuromorphic architectures, namely TrueNorth [5] and SpiNNaker [29]. The neural SSL model consisted of neurons smoothing gammatone-filtered input across frequency bands and time per hemisphere (S-T units), neurons differentially weighting input from S-T units of both hemispheres (W-S units) to compute ILD values, and a linear readout population with weights to W-S units that have been fit via regression to perform azimuthal SSL.

Neural models of SSL offer a way to study the complete range of functionality (adaptation to input statistics, top-down context-adaptivity, and learning) required to harness neuromorphic SSL computation for everyday real-world applications and usage in neuromorphic robotics. Thus, being able to bring one and the same SSL model to different neuromorphic platforms will help to pursue these goals further. To this end, we presented the first on-chip simulation results for the SSL model on both neuromorphic platforms.

To demonstrate the feasibility of our approach, we selected two neuromorphic architectures, TrueNorth and SpiNNaker. Not only do further architectures, such as Loihi [43] or BrainScaleS-2 [44] exist, the development in the field of neuromorphic computing is fast-paced so that new architectures, such as Loihi 2 [45] or SpiNNaker 2 [46], are expected to find broader research usage soon and further architectures will keep being developed. By having investigated implementations on TrueNorth and SpiNNaker, we expect that our procedure would also work considerably well for bringing SSL to other architectures as the frameworks and capabilities offered by TrueNorth and SpiNNaker are diametrically different—TrueNorth provides a specific neuron type and specific connectivity capabilities being optimized for high energy efficiency, while SpiNNaker offers broad flexibility for implementational decisions but offers fewer neurons and energy efficiency. Fittingly, the two have been termed “[...] good examples of the extremes one can take with digital hardware implementations”. [8], which makes them likewise good examples for the exemplification of bringing neural SSL to neuromorphic hardware by means of a generic mapping approach.

## Figures and Tables

**Figure 2 sensors-23-04451-f002:**
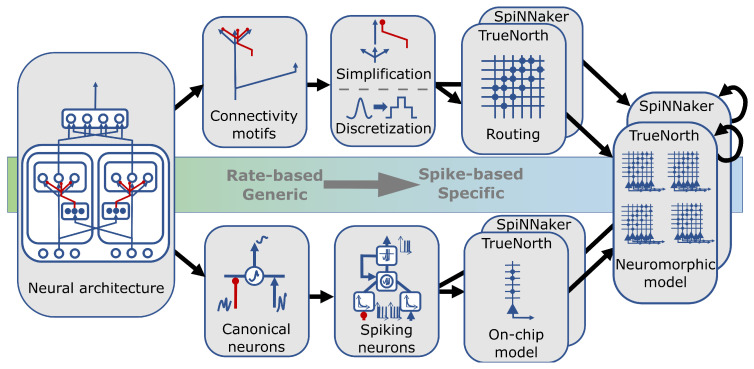
Overview of generic mapping from a rate-based neural architecture to specific neuromorphic platforms. Rate-based neural network architecture can be mapped onto specific neuromorphic hardware moving from hardware-agnostic to more and more hardware-specific mapping steps (from left to right). As part of the mapping procedure, the architecture needs to be treated on a network level (upper path) and a node level (lower path). Individual connections must be identified, simplified and discretized, and forwarded to a hardware-specific routing process to map onto a neuromorphic platform. Our proposed canonical neuron model [30] can be identified as components in the neural model, mapped from a rate-based to a spike-based model description, and then described and approximated in a hardware-specific fashion. Linking both, network-level and neuron-level descriptions, yields a complete neuromorphic model, which then finally can be fine-tuned on the respective hardware to best match the desired functional response properties.

**Figure 3 sensors-23-04451-f003:**
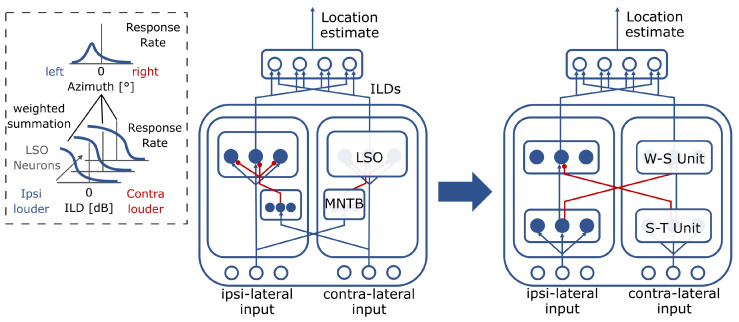
From the biologically inspired sound source localization model to a simplified version optimized for resource efficiency on neuromorphic hardware. As part of the proposed procedure to map the sound source localization model to neuromorphic hardware the multiplicity of kernel-based many-to-to-one connections is reduced to fewer many-to-many but more one-to-one connections while keeping the functionality intact. **Inset.** Lateral superior olive (LSO) neurons compute interaural level difference (ILD) values with tuning curve sensitivity being maximal for different azimuth angles for different neurons in the population. Linearly combining their responses allows us to estimate the input’s azimuth.

**Figure 4 sensors-23-04451-f004:**
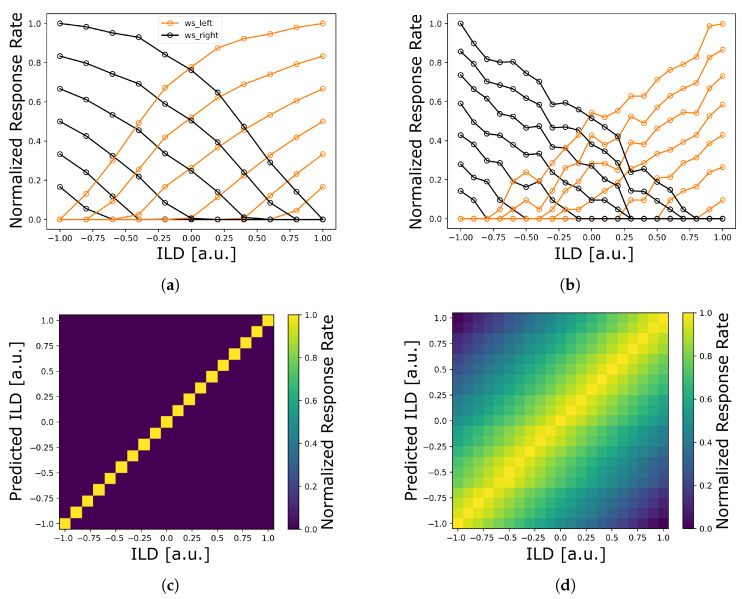
Auditory sound source localization results on neuromorphic platforms for synthetic input data. (**a**,**b**) Tuning curves of W-S units of a single frequency band for TrueNorth (**a**) and SpiNNaker (**b**). Curves display W-S unit activity for different azimuth angles of the input. Different curves correspond to different W-S units weighting ipsi- versus contra-lateral S-T unit input differently. (**c**,**d**) Readout characteristics (ordinate) based on W-S units of a single frequency for TrueNorth (**c**) and SpiNNaker (**d**) for a sweep over simulations with different interaural level difference (ILD) values (abscissa). Brighter values indicate neuron activation coding for higher evidence of the respective ILD being the correct one. TrueNorth plots are based on data from [21], which has been re-analyzed in the present study. See text for details.

**Figure 5 sensors-23-04451-f005:**
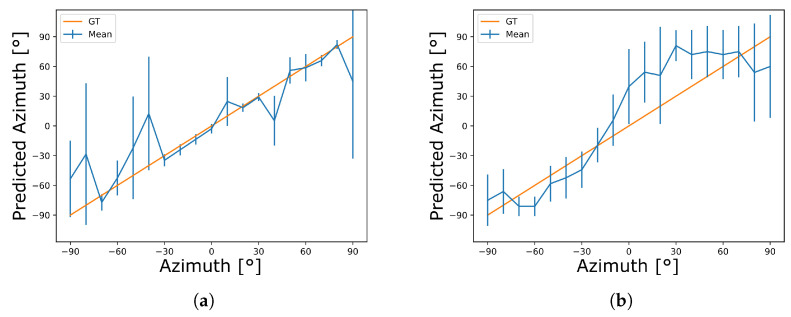
Auditory sound source localization results on TrueNorth (**a**) and SpiNNaker (**b**) for real-world stimuli. Ground truth (GT) data are replicated across both figures for comparison. Curves are means across the respective sound sources, error bars indicate ± one standard deviation.

**Table 1 sensors-23-04451-t001:** Overview of neuromorphic models for auditory sound source localization sorted by utilized hardware type (interaural time difference, ITD; interaural level difference, ILD; integrated circuit, IC; field-programmable gate array, FPGA; resistive random access memory, RRAM).

Method	Auditory Cue	Hardware	Input Data
Lazzaro et al. [16]	ITD	IC	On-chip, click stimuli
Glackin et al. [17]	ITD	FPGA	Ear canal measurements, sound-dampened chamber
Xu et al. [18]	ITD	FPGA	Real-world, reverberant
Escudero et al. [19]	ILD	FPGA	Real-world, pure tones
Schoepe et al. [20]	ITD	FPGA+SpiNNaker	Real-world, pure tones and human speech
Ours	ILD	SpiNNaker or TrueNorth [21]	Synthetic, or real-world sounds in sound-dampened chamber
Wang et al. [22]	ITD	RRAM	Proof-of-concept laboratory setup
Moro et al. [23]	ITD	RRAM	Proof-of-concept laboratory setup

## Data Availability

The SpiNNaker implementation, SpiNNaker results, as well as the real-world sound recordings filtered for usage with spiking networks are openly available at https://github.com/schmidDan/neuromrophic_auditory_ssl. The analyzed TrueNorth data underlie confidentiality clauses and are not publicly available.

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
