# Peer review of "Listen to the Brain–Auditory Sound Source Localization in Neuromorphic Computing Architectures"

_sensors, 2023, doi:10.3390/s23094451_

Round 1

Reviewer 1 Report

This paper discusses how conventional engineered sensory processing relies on fixed sampling at discrete time points, which results in redundancy in acquired data, leading to unnecessary consumption of resources. This study suggests that neuromorphic computing can mimic the biological retina and cochlea for vision and audition, respectively, by employing distributed, time-asynchronous sensory systems. This article then demonstrates how neuromorphic algorithms can be designed by modeling neural mechanisms known from the auditory system of mammals. The study presents model simulations for lateral auditory sound source localization (SSL) based on the interaural level difference (ILD) in the azimuthal plane. To operationalize the mapping of neural processing of auditory information onto different neuromorphic processors, the authors used two different brain-inspired hardware platforms, namely the IBM TrueNorth Neurosynaptic System and the SpiNNaker platform. The model simulations were successfully implemented on the different hardware designs of TrueNorth and SpiNNaker neuromorphic architectures, respectively, achieving energy-efficient real-time performance in SSL under realistic auditory conditions.

This paper needs extensive improvement to enhance its presentation. 

1) For a short communication, this abstract is very long. I suggest to make it shorten it.

2) The motivation and contributions of this work is not clear. The authors should clearly discuss that in the introduction. 

3) The authors should move the literature review to a another section, i.e., related works. Then, they need to discuss existing methods, identify their characteristics, advantages and limitations and summarize that on a table.

4) A conclusion should be added to the paper to discuss the important findings of this study, its limitations and future work. 

Author Response

Dear Reviewer,

thank you for commenting and providing valuable feedback on our manuscript.

The feedback we received from all reviewers greatly helped improving our paper. Hence, you may find that we performed substantial improvements on all parts of the manuscript. To name a few: We adapted the format to match the journal template, stated our motivations and contribution throughout the manuscript more concisely, added a complete new subsection towards reviewing relevant literature, added a subsection to explain model details, enhanced the presentation of our results, engaged more strongly with limitations and potential future work, and provide a separate conclusion section.

For a point-by-point coverage of your comments we kindly refer you to the attached document.

Reviewer 2 Report

The article examines the implementation of an auditory sound source localization model on TrueNorth and SpiNNaker neuromorphic hardware. Two experiments were conducted: the first aimed to determine the model's ability to accurately encode Interaural Level Differences (ILDs), while the second evaluated the model's performance on real-world sounds.   The authors need to clarify a few questions:

1.  Although the study includes both synthetic and real-world data, the synthetic data is restricted to a single frequency band and fails to capture the intricacies and heterogeneity of real-world sounds. Furthermore, the real-world data is constrained to a narrow range of natural sounds and captured in a sound-dampened chamber, which may not fully reflect the diversity of natural listening conditions.

2. The article only reports on one experiment for each implementation, which makes it difficult to draw firm conclusions about the performance of the models.

3.   It is not clear if the article discusses the limitations of using neuromorphic hardware for sound source localization, such as the computational cost and power consumption of the hardware and comparing it with existing approaches/hardware.

Author Response

Dear Reviewer,

thank you for commenting and providing valuable feedback on our manuscript.

The feedback we received from all reviewers greatly helped improving our paper. Hence, you may find that we performed substantial improvements on all parts of the manuscript. To name a few: We adapted the format to match the journal template, stated our motivations and contribution throughout the manuscript more concisely, added a complete new subsection towards reviewing relevant literature, added a subsection to explain model details, enhanced the presentation of our results, engaged more strongly with limitations and potential future work, and provide a separate conclusion section.

We aimed at clarifying your questions within the attached reply, as well as incorporating respective improvements within our manuscript. For a point-by-point coverage of your comments we kindly refer you to the attached document.

Reviewer 3 Report

The paper is relevant and well written. Just tiny revision.

Your abstract seems well organized, but the length is too long and needs to be reduced, and the results part in the abstract lacks data, not recommended to use text only.

General comment on the Introduction section: my main suggestion is to shorten the introduction that is a bit too long and to make a deeper analysis of the most recent literature. Besides, some knowledge and methodological backgrounds were not presented in the introduction and methodology but with results.

The format of the paper needs to be modified according to the template of the journal. The secondary title in the text needs to be numbered.

Conclusions: Further focus on your results/findings.

I have no strong plagiarism checker and you should do that.

Author Response

Dear Reviewer,

thank you for kindly regarding our manuscript and providing valuable feedback in terms of your comments.

The feedback we received from all reviewers greatly helped improving our paper. Hence, you may find that we performed substantial improvements on all parts of the manuscript. To name a few: We adapted the format to match the journal template, stated our motivations and contribution throughout the manuscript more concisely, added a complete new subsection towards reviewing relevant literature, added a subsection to explain model details, enhanced the presentation of our results, engaged more strongly with limitations and potential future work, and provide a separate conclusion section.

For a point-by-point coverage of your comments we kindly refer you to the attached document.

Reviewer 4 Report

I am sorry that the positioning of this article is not consistent with what I understand. This article is mainly about designing a device that simulates human fuzzy sound source localization, while I think this paper is about precise localization on machines. Therefore, my comments are for reference only.

1. The description of mathematical model is insufficient, and the mathematical process of how to obtain position estimates from parameters such as ILD/ITD is not explained.

2. The final simulation results have no example of positioning results, which is somewhat deviated from the title of the paper. At least, the author can provide an example with source-device topology,and present the estimated angle and position error.

3. Because in broadband sound signals, it is impossible for two ears to meet the half wavelength spacing requirement, accurate direction estimation is indeed impossible. However, the principle of ILD for positioning needs to be clarified. Generally, the final position of this kind of differential parameter is a curve, unless more sensing nodes are introduced.

Author Response

Dear Reviewer,

thank you for commenting on our manuscript and providing valuable feedback.

The feedback we received from all reviewers greatly helped improving our paper. Hence, you may find that we performed substantial improvements on all parts of the manuscript. To name a few: We adapted the format to match the journal template, stated our motivations and contribution throughout the manuscript more concisely, added a complete new subsection towards reviewing relevant literature, added a subsection to explain model details, enhanced the presentation of our results, engaged more strongly with limitations and potential future work, and provide a separate conclusion section.

We especially welcome your comments from a different perspective on our manuscript and are confident, that it led to a great improvement in how we can present our approach to a broader audience. For a point-by-point coverage of your comments we kindly refer you to the attached document.

Round 2

Reviewer 1 Report

The authors have addressed all my comments, I have no further suggestions. I am happy with the actual version.